# Misreporting contraceptive use and the association of peak study progestin levels with weight and BMI among women randomized to the progestin-only injectable contraceptives DMPA-IM and NET-EN

Chanel Avenant[1], Alexis J. Bick[1], Salndave B. Skosana[1†], Sigcinile Dlamini[1], Yusentha Balakrishna[2], Johnson Mosoko Moliki[1], Mandisa Singata-Madliki[3], G. Justus Hofmeyr[3,4,5], Jenni Smit[6], Mags Beksinska[6], Ivana Beesham[6], Ishen Seocharan[2], Joanne Batting[3], Pai-Lien Chen[7], Karl-Heinz Storbeck[8], Donita Africander[8], Janet P. Hapgood[1,9]*

1 Department of Molecular and Cell Biology, University of Cape Town, Cape Town, South Africa, 2 Biostatistics Research Unit, South African Medical Research Council, Durban, South Africa, 3 Effective Care Research Unit, Eastern Cape Department of Health, Universities of the Witwatersrand and Fort Hare, East London, South Africa, 4 Walter Sisulu University, East London, South Africa, 5 Department of Obstetrics and Gynecology, University of Botswana, Gaborone, Botswana, 6 Wits MRU (MatCH Research Unit), Department of Obstetrics and Gynecology, Faculty of Health Sciences, University of the Witwatersrand, Durban, South Africa, 7 Family Health International (FHI) 360, Durham, North Carolina, United States of America, 8 Department of Biochemistry, Stellenbosch University, Stellenbosch, South Africa, 9 Institute of Infectious Disease and Molecular Medicine, University of Cape Town, Cape Town, South Africa

† Deceased.
* janet.hapgood@uct.ac.za

⊙ OPEN ACCESS

## Abstract

Progestin-only injectable contraceptives, mainly depo-medroxyprogesterone acetate intramuscular (DMPA-IM), are the most widely used contraceptive methods in sub-Saharan Africa. Insufficient robust data on their relative side-effects and serum concentrations limit understanding of reported outcomes in contraception trials. The WHICH clinical trial randomized HIV-negative women to DMPA-IM (n = 262) or norethisterone enanthate (NET-EN) (n = 259) at two South African sites between 2018–2019. We measured serum concentrations of study and non-study progestins at initiation (D0) and peak serum levels, one week after the 24-week injection [25 weeks (25W)], (n = 435) and investigated associations between study progestin levels, and BMI and weight of participants. Peak median serum concentrations were 6.59 (IQR 4.80; 8.70) nM for medroxyprogesterone (MPA) (n = 161) and 13.6 (IQR 9.01; 19.0) nM for norethisterone (NET) (n = 155). MPA was the most commonly quantifiable non-study progestin at D0 in both arms (54%) and at 25W in the NET-EN arm (27%), followed by NET at D0 in both arms (29%) and at 25W in the DMPA-IM arm (19%). Levonorgestrel was quantifiable in both arms [D0 (6.9%); 25W (3.4%)], while other progestins were quantifiable in ≤ 14 participants. Significant negative time-varying associations were detected between MPA and NET concentrations and weight and BMI in both contraceptive arms and a significant increase was detected for peak serum progestin

**Data Availability Statement:** We have uploaded the raw data to the South African Medical Research Council (SAMRC) database and it can be found under the link: https://medat.samrc.ac.za/index.php/catalog/52.

**Funding:** This work was supported by the U.S. National Institutes of Health and South African Medical Research Council through its U.S.-SA Program for Collaborative Biomedical Research (R01HD083026 [NICHD & NIAID] and R01AI152118 [NIAID]) to J.P.H. (https://www.nih.gov & https://www.samrc.ac.za), and a UCT Vice Chancellor's Advancing Womxn award to J.P.H (https://uct.ac.za). The clinical trial was funded by the South African Medical Research Council Grants, Innovation and Product Development (SAMRC N/A grant number) grant to M.S-M (https://www.samrc.ac.za/innovation/grants-innovation-and-product-development). The funders had no role in study design, data collection and analysis, decision to publish, or preparation of the manuscript.

**Competing interests:** The authors have declared that no competing interests exist.

concentrations for normal weight versus obese women. Contraceptive-related reported outcomes are likely confounded by MPA, more so than NET, with reported DMPA-IM effects likely underestimated, at sites where DMPA-IM is widely used, due to misreporting of contraceptive use before and during trials, and 'tail' effects of DMPA-IM use more than six months before trial enrolment. Peak serum levels of MPA and NET are negatively associated with BMI and weight, suggesting another source of variability between trial outcomes and a potential increase in side-effects for normal weight versus overweight and obese women.

**Trail registration:** The clinical trial was registered with the Pan African Clinical Trials Registry (PACTR 202009758229976).

## Introduction

Injectable progestin-only contraceptives are the most used contraceptive methods in sub-Saharan Africa [1]. Depo-medroxyprogesterone acetate (DMPA) is a three-monthly progestin-only intramuscular (IM) injectable contraceptive containing 150 mg MPA and is the most commonly used injectable contraceptive method in sub-Saharan Africa and South Africa [2]. The second most common injectable contraceptive is norethisterone enanthate (NET-EN), a two-monthly contraceptive that contains 200 mg of the progestin NET and is mainly used in South Africa [2]. Unique to injectable contraceptives is their pharmacokinetic profile which shows an initial peak concentration within the first few weeks post-injection, followed by a decline to steady-state levels that maintain contraceptive efficacy until the next dose 2–3 months later [3, 4].

Previous studies have reported large inter-individual and inter-study variations in serum MPA and NET levels, with differences in reported peak serum levels of 38- to 48-fold between studies (reviewed in [5]). MPA in serum of DMPA-IM users reportedly reaches a peak of 2.6–100 nM within the first few weeks post-injection, while peak NET levels reach 2.4–117 nM (reviewed in [5–7]). However, many of the published studies of peak concentrations had small sample sizes and used quantification methods such as radioimmunoassay without prior extraction (reviewed in [7]). More recent clinical studies, although using high-throughput and highly sensitive liquid chromatography-mass spectrometry (LC-MS) methods, may be confounded by: unequal numbers of participants using DMPA-IM and NET-EN, comparing samples from non-randomized trials or inter-study comparisons, differences in intrinsic participant demographic characteristics, using sampling time frames that do not correlate with peak levels, and using different progestin quantification methods [5, 7]. Our detailed review of available data suggests that peak serum levels of MPA and NET are reached at about one week post-injection [5–7].

Determining the concentrations of serum progestins is important for several reasons. Progestins may elicit dose-dependent biological side-effects such as increased risk of HIV and other sexually transmitted infections, decreased bone density and weight change [8, 9]. Systematic reviews suggest a 32–40% lower risk of HIV acquisition among participants using NET-EN versus DMPA-IM [4, 10–12], stimulating interest and urgency in obtaining data on the relative effects of DMPA-IM and NET-EN on health outcomes. Some studies conducted among users of etonogestrel (ETG) or levonorgestrel (LNG) implants suggest a negative association between body mass index (BMI) and serum progestin levels [13, 14]. One study showed that months of ETG implant use was significantly associated with decreased serum ETG concentrations [15]. However, such associations were not observed in other implant studies [16, 17] nor in studies among DMPA-IM or NET-EN users [5, 18–22]. These studies with DMPA-IM and NET-EN

may have lacked sufficient power to detect such associations. The change in injectable contraceptive concentrations over time implies that side-effects may be more pronounced during peak serum concentrations compared to lower concentrations. Such concentrations may be lower in obese or overweight women [23], which is prevalent in some regions of sub-Saharan Africa and South Africa. It has been shown that obesity reduces sensitivity to MPA when used in cancer treatment [24, 25], while another study reported that abnormal uterine bleeding in DMPA-IM users is lower in overweight and obese women than in normal weight women [26], consistent with our hypothesis. Furthermore, the large inter-individual variation in serum contraceptive levels suggests that side-effects may be greater at peak levels for some women compared to others.

Differential side-effects between MPA and NET are likely mediated by their different affinities for and activities via members of the steroid receptor family [8, 27]. Both MPA and NET bind to and activate the progesterone receptor (PR) and androgen receptor (AR), whereas MPA, but not NET, activates the glucocorticoid receptor (GR) [8, 27]. Steroid receptors are ligand-activated transcription factors: the PR predominantly regulates reproductive function while the GR is a key regulator of metabolic, anti-inflammatory and immune responses [8, 27, 28]. Therefore, side-effects may be mediated by the PR and AR in NET-EN users and by the GR and/or PR and/or AR in DMPA-IM users, to variable degrees, depending on the progestin concentrations over time.

It is becoming clear that objective quantification of serum progestin levels is necessary to validate unreliable self-reporting of contraceptive use within and prior to the trial enrolment [29–37]. To date, the majority of contraceptive studies, including HIV incidence studies, have investigated DMPA-IM, while studies in NET-EN users are limited and others report effects of injectable contraceptives together without disaggregating them [4, 10, 11, 38]. Nevertheless, quantification of MPA in DMPA-IM users is now becoming a routine procedure, whereas studies measuring NET in NET-EN users are limited [31, 33, 39]. Only a few head-to-head studies have compared MPA and NET serum concentrations [18, 33, 40]. However, differences in sampling times, sample sizes, quantification methodology and demographics between these studies likely contribute to the large variation in NET and MPA levels reported (reviewed in [5, 7]). Importantly, samples from a Zimbabwean trial, the Zim-CHIC cohort, were taken at 30-, 90- and 180-days post-injection [33], and thus the earliest time point of one month after injection is likely past the peak for serum progestin levels.

We therefore sought to quantify MPA and NET concentrations in a highly powered randomized trial without several of the abovementioned confounding factors. We report here for the first time, the baseline and peak serum concentrations of MPA and NET in a large cohort of 435 South African women randomized equally to either DMPA-IM or NET-EN, from the Women's Health, Injectable Contraceptive and HIV (WHICH) clinical trial. Samples were taken at study enrolment, prior to the first injection (baseline, D0) and one-week after the 24 weeks (24W) contraceptive injection, i.e. after three doses of DMPA-IM and four doses of NET-EN (at 25 weeks or 25W). We performed ultra-high performance liquid chromatography tandem mass spectrometry (UHPLC-MS/MS) to investigate the concentrations of MPA and NET as well as other potentially confounding progestins that are commonly used in alternative contraceptive methods in South Africa [2]. We also investigated associations between MPA and NET concentrations, and BMI and weight.

## Methods

### Primary clinical trial methods and ethics

This study is a secondary study of the parent study i.e. the open-label randomized WHICH clinical trial. The primary aims of the WHICH trial were to evaluate estradiol levels and

menstrual, psychological and behavioral measures relevant to HIV risk. The primary results will be reported elsewhere and are under review. A summarized protocol for the trial is available at https://pactr.samrc.ac.za/TrialDisplay.aspx?TrialID=6073. The study was registered with the Pan African Clinical Trials Registry number PACTR 202009758229976 (https://pactr.samrc.ac.za/Search.aspx). In 2020 the authors discovered that the initial online trial registration on the Pan African Clinical Trials Registry website had not been logged onto the system, due to missing information on individual participant data sharing. The original information plus individual participant data sharing statement were re-entered and approved on 1 September 2020. The authors confirm that all ongoing and related trials for this drug/intervention are registered.

This trial, conducted at two sites in South Africa from 2018 to 2019, equally randomized HIV-negative women aged 18–40 years to 150 mg DMPA-IM 12-weekly or 200 mg NET-EN intramuscular eight-weekly. Women seeking contraception were recruited at the East London and Mdantsane public health clinics and hospitals (Frere and Cecilia Makiwane Hospitals) (ECRU), South Africa (331 participants), and the research site of the University of the Witwatersrand MatCH Research Unit (MRU), based in Durban, KwaZulu-Natal, South Africa (189 participants). Exclusion criteria included participants who had received DMPA-IM in the previous six months or NET-EN in the previous four months via self-report, were living with HIV, or were using or intending to use medication which might have interfered with biological measurements such as steroids or drugs affecting renal function, such as pre-exposure prophylactic drugs (PrEP). Participants were recruited and followed from 5 November 2018 to 30 November 2019. We screened 546 and randomized 521 women to DMPA-IM (262) and NET-EN (259). A total of 86.9% (n = 453) completed a 25-week study visit with a similar number completing in both method groups.

Blood samples were collected at study D0 and at 25W (about seven days after the 24W progestin injection), and serum was separated and stored at -80˚C. Ethical approval for the trial was obtained from the Faculty of Health Sciences Human Research Ethics Committee (FHS HREC, M180528) of the University of Witwatersrand Research Ethics Committee, and from the East London Hospital Institutional Ethics Committee. Permission to conduct the trial was obtained from the Provincial Departments of Health of the Eastern Cape and KwaZulu-Natal. BMI and weight were measured by trained staff at D0 and 24W. Weight/BMI categories were assigned according to [41]. BMI was calculated by dividing weight in kilograms by the height in metres squared ($kg/m^2$). Ethical approval for the secondary study conducted at the University of Cape Town (UCT), some of the results of which are reported in this manuscript, was obtained from the UCT Faculty of Health Sciences Human Research Ethics Committee (HREC REF no. 664/2018). All women provided informed, written consent to authorize study participation and storage of samples. This study adhered to the ethical principles outlined in the Declaration of Helsinki (World Medical Association, 2011) and the Constitution of the Republic of South Africa (Bill of Rights). For additional details on the trial methodology see the S1 Protocol. The authors did not have access to information that could identify individual participants during or after data collection.

## Progestin measurements

These measurements were performed between 2 January 2020 and 31 December 2022. Progestins were measured by UHPLC-MS/MS on stored samples from 435 WHICH study participants at D0 and 25W as described in S2 Protocol. Retention times, mass transitions and comprehensive method validation are reported in S1 and S2 Tables. MPA, NET and nestorone (NES) had a lower limit of quantification (LLOQ) of 0.05 ng/mL (MPA 0.129 nM; NET 0.168

nM; NES 0.135 nM), followed by levonorgestrel (LNG) and ETG with a LLOQ of 0.1 ng/mL (LNG 0.320 nM; ETG 0.308 nM), while the LLOQ for gestodene (GES) was 0.5 ng/mL (1.61 nM) (S3 Table). The upper limit of quantification (ULOQ) was 50 ng/mL (the highest concentration used) for all progestins, which is well above the highest concentration found in the serum samples (S3 Table).

## Data analysis

UHPLC-MS/MS data collection and analysis were performed using MassLynx 4.2 (Waters Corporation). The ratio of the analyte peak area to internal standard peak area was determined for all the calibration curve samples, internal quality controls (IQCs) and serum samples. While the peak area of MPA-d6 or NET-d6 was used to quantify MPA or NET, respectively, the mean peak area of MPA-d6 and NET-d6 was used for the quantification of LNG, ETG, NES and GES.

The analysis of unimputed data used only observed concentrations higher than the LLOQ. In other words, only the participants with quantifiable levels are included in this analysis. For imputed data, values between the LLOQ and the limit of detection (LOD) were assigned 0.5 x LLOQ, while values lower than LOD were assigned as 0.000. For the imputed data, all participants were included in the data analysis.

We performed modified intention-to-treat (mITT) and per protocol (PP) analyses (PP1 and PP2). For the PP1 analysis, to obtain the most robust study progestin peak serum values at 25W, we excluded participants that did not receive their 24W injection (DMPA-IM n = 7; NET-EN n = 3) and those that had a concentration of non-study progestins greater than its LLOQ at 25W (DMPA-IM n = 47; NET-EN n = 62). For the PP2 analysis, to investigate associations between progestin concentrations, weight and BMI, we excluded participants that did not receive their 24W injection (DMPA-IM n = 7; NET-EN n = 3) and those with non-study serum progestin concentrations greater than 1.5 nM, at both D0 and 25W (DMPA-IM n = 27; NET-EN n = 66).

Differences in baseline characteristics between arms were assessed using Pearson's chi-squared or Fisher's exact test, where applicable, or a two-sample t-test (for differences in age) and a Wilcoxon rank-sum test (for differences in weight and BMI).

Progestin results were analysed using GraphPad Prism 9.31 from GraphPad Software, Inc. (La Jolla California, USA) and SAS 9.4 (SAS Institute Inc., Cary, USA). All data in the main text, unless otherwise indicated, are expressed as median with interquartile range (IQR). For some tables, data are expressed as median with minimum and maximum values, to highlight outliers. Means with minimum and maximum values are also given for some of the data, where thought to be informative (S4, S5 and S7 Tables). All the progestin data were not normally distributed, as determined by Shapiro-Wilk test, and hence only non-parametric tests were performed to obtain crude p-values. The Wilcoxon matched-pairs signed rank test was used for within group/arm analysis (within the DMPA-IM arm comparing D0 to 25W, or within the NET-EN arm comparing D0 to 25W), while the Mann Whitney test was performed when comparing between groups/arms (DMPA-IM D0 vs NET-EN D0, or DMPA-IM 25W vs NET-EN 25W), as well as for the absolute change (25W –D0) between groups/arms. Levene's test was used to assess the similarity of the variations in progestin concentrations between DMPA-IM and NET-EN groups [42]. To assess whether study location influenced progestin comparison results, generalized linear models [43] were applied to obtain site-adjusted p-values using progestin data transformed by Box-Cox power transformation approaches [44]. The same approach was used to evaluate time varying associations between progestin concentrations and demographic factors based on D0 and 25W time points. Associations between 25W

serum MPA or NET concentrations and either D0 or 24W weight or BMI were investigated by non-parametric Spearman correlation analysis. Furthermore, Dunn's multiple comparison test was used to compare the median peak serum MPA and NET concentrations among women's weight categories based on their BMIs.

## Results

### Whole cohort peak serum and baseline MPA and NET concentrations (mITT analysis)

Among 521 participants enrolled, results were available for 435 (83%) participants both at baseline and at peak serum (Fig 1). The excluded participants included 11 (2%) that became HIV positive, 6 (1%) that became pregnant, 65 (12%) that were lost to follow up (i.e. those that did not provide a 25W blood sample) and 3 participants that had missing progestin results, due to technical problems with the sample. The trial profile is shown in Fig 1.

Baseline data is shown in Table 1. There was no difference in mean age (25 years), ethnicity or previous use of method (DMPA-IM or NET-EN) between the two arms. The women assigned to DMPA-IM weighed significantly more and had a significantly higher BMI than those assigned to the NET-EN arm in the whole cohort for the mITT analysis (Table 1).

We measured MPA and NET concentrations by UHPLC-MS/MS in serum from all the available matched D0 and 25W samples from the WHICH clinical trial cohort (215 and 220 women for the DMPA-IM and NET-EN arms, respectively) and performed a mITT analysis on this data. The median peak serum concentration for MPA in the DMPA-IM arm was 6.59 nM (2.55 ng/mL) (IQR 4.80–8.70 nM) and for NET in the NET-EN arm was 13.6 nM (4.06 ng/mL) (IQR 9.01–19.0 nM) (Table 2). While the mean peak serum concentrations for MPA and NET in the respective arms were only slightly higher (7.03 nM MPA in the DMPA-IM arm; 14.9 nM NET in the NET-EN arm), there were individuals with 2.9- and 3.6-fold higher concentrations of MPA or NET, respectively, than the means for MPA (20.4 nM) and NET (52.9 nM) (S4 Table and Fig 2). No significant differences were detected between arms in the median or mean concentrations of MPA or NET levels at D0 (Table 2 and S4 Table). At D0 the median concentration of MPA in the DMPA-IM arm was 0.215 nM (0.00–1.39 nM), while for NET in the NET-EN arm it was 0.00 nM (IQR 0.00–0.149 nM). At D0, more individuals had quantifiable levels of MPA (115 (53.5%) in the DMPA-IM arm and 121 (55.0%) in the NET-EN arm) compared to NET (69 (32.1%) in the DMPA-IM arm and 55 (25.0%) in the NET-EN arm) (Fig 2 and Table 3). Nevertheless, no significant differences for unadjusted or site-adjusted values were detected between arms in the median or mean concentrations of MPA or NET levels at D0 (Table 2 and S4 Table). In addition, even at 25W, several women still had quantifiable MPA in the NET-EN arm (60 (27.3%)), while fewer had quantifiable NET in the DMPA-IM arm (40 (18.6%)) (Table 3). Some individuals in the DMPA-IM arm had high levels (up to ~ 20 nM) of NET at 25W (Fig 2 and Table 3), while there were individuals in the NET-EN arm that had high levels of MPA (up to ~ 20 nM) at 25W.

The mean and median time (days) between the final contraceptive injection at 24W and when the sample (blood, serum for UHPLC-MS/MS) was taken, was 7.0 days for both contraceptive arms in both the mITT and PP1 analysis (S5 Table and S1 Fig). Spearman correlation analysis showed a significant negative correlation between MPA (p = 0.0381) and NET (p = 0.0002) concentrations at 25W and the number of days since the 24W injection with DMPA-IM and NET-EN, respectively (S1 Fig). In addition, significant negative time-varying associations were detected between the change in MPA (p<0.001) and NET (p<0.001) concentrations and the number of days since the 24W injection with DMPA-IM and NET-EN, respectively (data not shown).

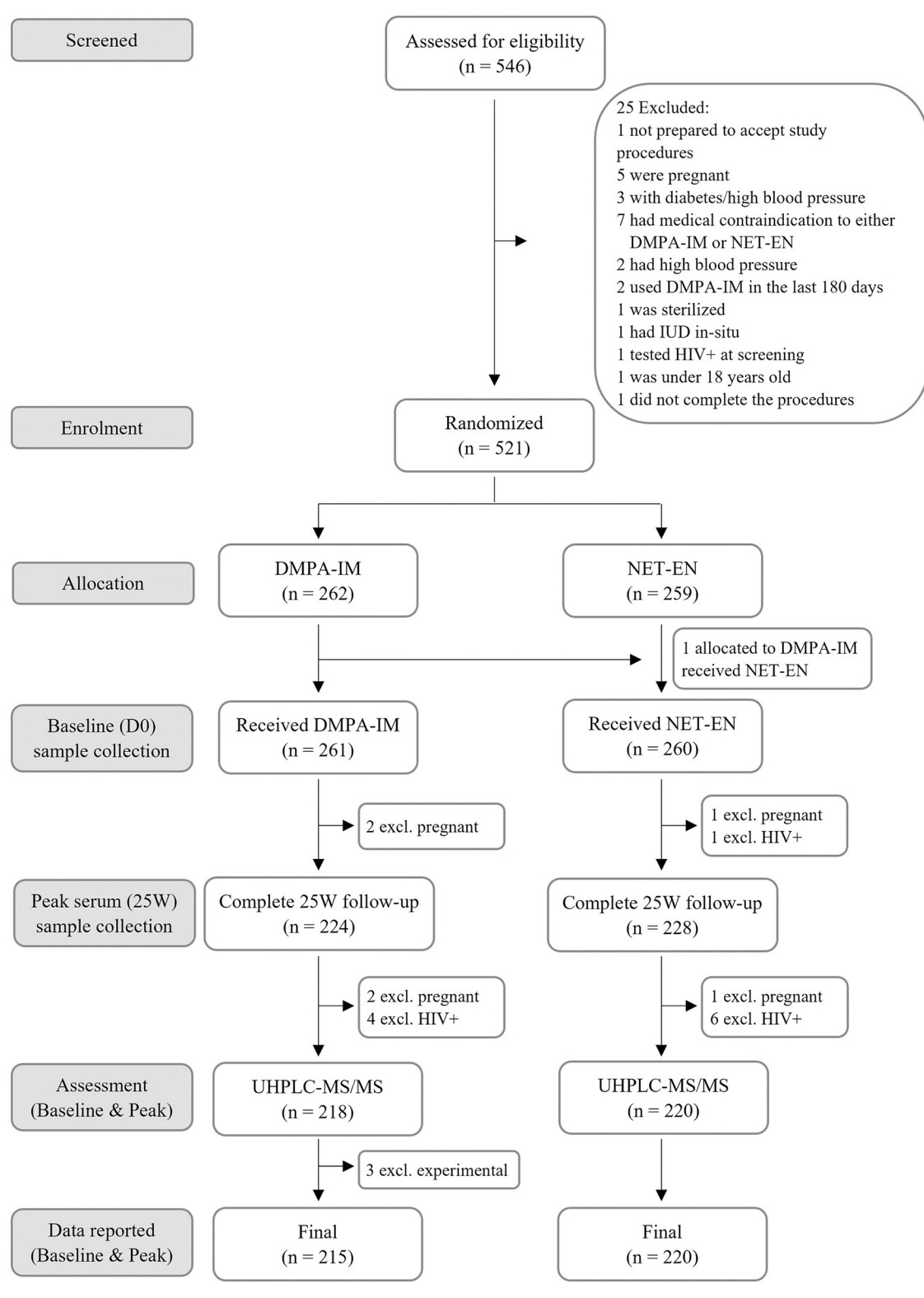

**Fig 1. Trial profile.**

**Table 1. Baseline characteristics of women by randomisation method, for whole cohort (mITT) and subgroup (PP1).**

| | mITT | | | | | | PP1 | | | | |
| | DMPA-IM | | NET-EN | | DMPA-IM vs NET-EN | | DMPA-IM | | NET-EN | | DMPA-IM vs NET-EN |
| | | n | | n | p-value* | | | n | | n | p-value* |
|---|---|---|---|---|---|---|---|---|---|---|---|
| Age, years: mean (SD) | 25 (4.8) | 217 | 24.9 (4.7) | 219 | 0.880 | | 25.2 (4.7) | 161 | 25.1 (4.6) | 155 | 0.872 |
| Weight, kgs: median (IQR) | 75 (60.1; 89.5) | 217 | 69 (58.2; 84.3) | 218 | **0.025** | | 75.2 (59.3; 90.0) | 161 | 70.9 (58.5; 87) | 155 | 0.209 |
| BMI: median (IQR) | 29.2 (24.3; 35.6) | 217 | 27.4 (23.6; 33.7) | 218 | **0.032** | | 29.4 (24.1; 36.5) | 161 | 27.6 (23.2; 35.3) | 155 | 0.118 |
| BMI category: n-value (%) | | 217 | | 218 | 0.140 | | | 161 | | 155 | 0.163 |
| Underweight | 2 (0.9) | | 6 (2.8) | | | | 1 (0.6) | | 5 (3.2) | | |
| Normal | 61 (28.1) | | 77 (35.3) | | | | 45 (27.9) | | 54 (34.8) | | |
| Overweight | 51 (23.5) | | 50 (22.9) | | | | 36 (22.4) | | 32 (20.7) | | |
| Obese | 103 (47.5) | | 85 (39.0) | | | | 79 (49.1) | | 64 (41.3) | | |
| Ethnicity: n-value (%) | | 217 | | 219 | 0.160 | | | 161 | | 155 | 0.195 |
| Xhosa | 145 (66.8) | | 148 (67.6) | | | | 111 (68.9) | | 108 (69.7) | | |
| Zulu | 67 (30.9) | | 71 (32.4) | | | | 46 (28.6) | | 47 (30.3) | | |
| Mixed race | 1 (0.5) | | 0 (0.0) | | | | 0 (0.0) | | 0 (0.0) | | |
| Other African ethnicity | 4 (1.8) | | 0 (0.0) | | | | 4 (2.5) | | 0 (0.0) | | |
| Previous use of method[#] | | | | | | | | | | | |
| DMPA-IM | 161 (74.2) | 217 | 160 (73.1) | 219 | 0.790 | | 125 (77.6) | 161 | 113 (72.9) | 155 | 0.329 |
| NET-EN | 69 (31.8) | 217 | 65 (29.7) | 219 | 0.630 | | 41 (25.5) | 161 | 46 (29.7) | 155 | 0.402 |

[#] Prior to exclusion period, numbers given are for those that responded, and in brackets are % of those that responded. Age compared using a two-sample t-test. Weight and BMI compared using Wilcoxon rank-sum test. All other characteristics compared using Pearson's chi-squared test or Fisher's exact test, where appropriate.

## Concentrations of non-study progestins

We also determined the concentration of LNG, ETG, NES and GES. The most commonly quantifiable non-study progestin detected at D0 was LNG (30 women), followed by NES and ETG (8 and 5 women, respectively), with GES quantifiable in only 2 women (Table 3). Non-

**Table 2. Analysis of MPA and NET imputed concentrations for whole cohort (mITT).**

| | DMPA-IM | | NET-EN | | DMPA-IM vs NET-EN |
| | Median (IQR) | n | Median (IQR) | n | p-value* |
|---|---|---|---|---|---|
| MPA (nM) | | | | | |
| D0 | 0.215 (0.00; 1.39) | 215 | 0.211 (0.00; 1.72) | 220 | 0.897 |
| 25W | 6.59 (4.80; 8.70) | 215 | 0.00 (0.00; 0.00) | 220 | **<0.0001** |
| Change (25W - D0) | 5.63 (3.69; 7.96) | | -0.120 (-1.33; 0.00) | | **<0.0001** |
| Change p-value* | **<0.0001** | | **<0.0001** | | |
| NET (nM) | | | | | |
| D0 | 0.00 (0.00; 0.308) | 215 | 0.00 (0.00; 0.149) | 220 | 0.149 |
| 25W | 0.00 (0.00; 0.0840) | 215 | 13.6 (9.01; 19.0) | 220 | **<0.0001** |
| Change (25W - D0) | 0.00 (-0.194; 0.00) | | 13.2 (8.80; 18.9) | | **<0.0001** |
| Change p-value* | **<0.0001** | | **<0.0001** | | |

*Unadjusted; IQR (25th and 75th Percentile); Within group analysis—Wilcoxon matched-pairs signed rank test; Between groups (D0 vs D0 and 25W vs 25W) and Change (absolute 25W –D0)—Mann Whitney test.

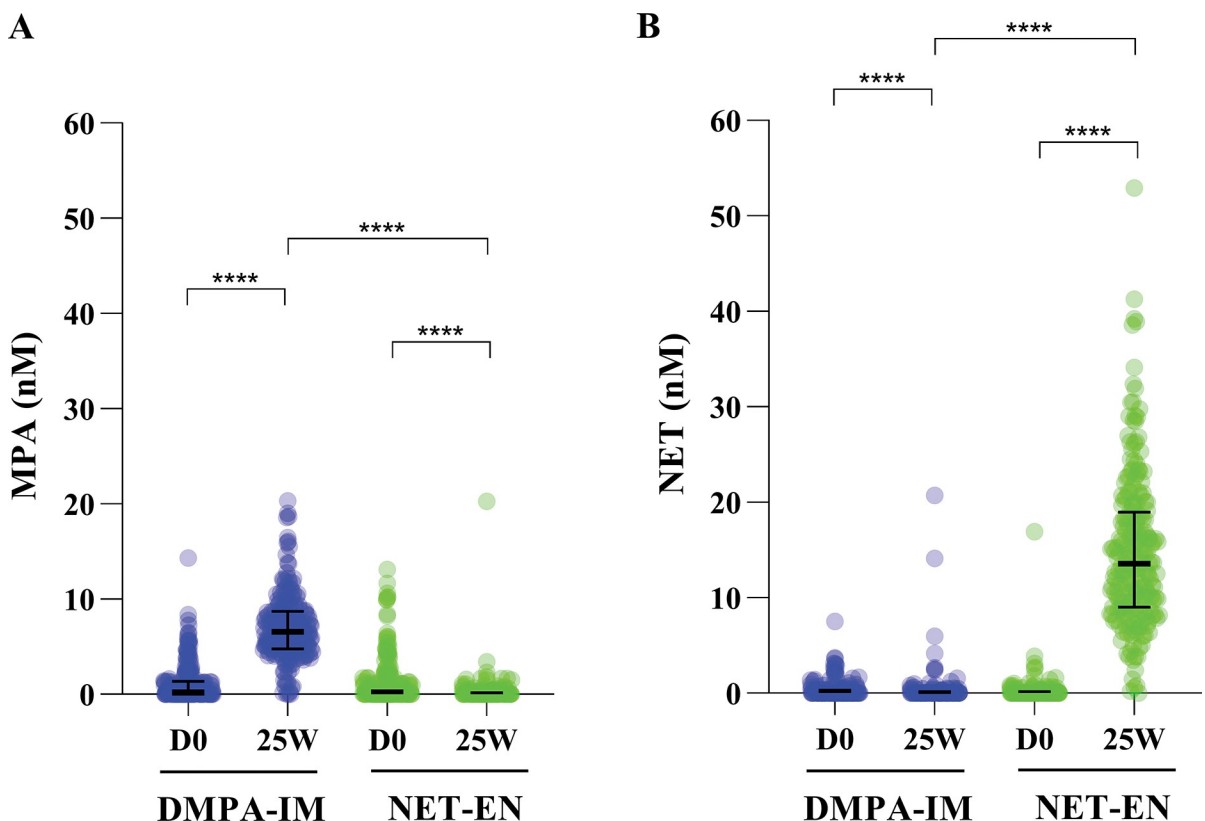

**Fig 2.** MPA (A) and NET (B) imputed concentrations for whole cohort (mITT). Graphs indicate median with interquartile range (IQR). Within group analysis—Wilcoxon matched-pairs signed rank test; Between groups (MPA D0 vs NET D0 and MPA 25W vs NET 25W)—Mann Whitney test. Significant differences are indicated by asterisks where *** and **** represent p<0.001 and p<0.0001, respectively.

study progestins were also detected at 25W, but for fewer women than at D0 for LNG and ETG, and the same number or more for GES and NES, respectively (Table 3). In those individuals for whom quantifiable non-study progestins were detected in the whole cohort, the median concentrations at D0 ranged from 0.260 nM to 2.16 nM, with some individuals having concentrations as high as 6.24 nM (e.g. for LNG) (Table 3). At 25W, the median concentration of quantifiable non-study progestins in individuals for whom quantifiable non-study progestins were detected in the whole cohort ranged from 0.310 nM to 4.40 nM, with some individuals having concentrations as high as 7.08 nM (e.g. for LNG). When considering the whole cohort, no statistically significant differences were detected between the median concentrations for any of these progestins at D0 or 25W between the DMPA-IM and NET-EN arms, with the exception of NES (data not shown). The median concentrations of LNG, ETG, NES and GES at D0 and 25W were 0.00 nM in both arms (data not shown). We did not detect any differences in the percentage (%) non-study progestins between the trial sites (S6 Table).

## Subgroup peak serum MPA and NET concentrations

Given that non-study progestins were detected in a large proportion of the women in the mITT group at 25W, and that several women had low quantifiable study progestins at 25W, we performed a PP analysis on a subgroup of the whole cohort to reduce these potential confounding effects on peak serum levels at 25W. In this PP1 analysis, we excluded all

**Table 3. Analysis of quantifiable unimputed concentrations of all progestins for whole cohort (mITT).**

| | DMPA-IM (n = 215) | | | NET-EN (n = 220) | | | Both arms (n = 435) | | |
|---|---|---|---|---|---|---|---|---|---|
| | n* | %# | Median (Min; Max) nM | n* | %# | Median (Min; Max) nM | n* | %# | Median (Min; Max) nM |
| **MPA** | | | | | | | | | |
| D0 | 115 | 53.5 | 1.30 (0.129; 14.3) | 121 | 55.0 | 1.27 (0.142; 13.1) | 236 | 54.3 | 1.28 (0.129; 14.3) |
| 25W | 212 | 98.6 | 6.72 (0.168; 20.4) | 60 | 27.3 | 0.413 (0.129; 20.3) | | | ND |
| **NET** | | | | | | | | | |
| D0 | 69 | 32.1 | 0.516 (0.174; 7.51) | 55 | 25.0 | 0.352 (0.171; 16.9) | 124 | 28.5 | 0.413 (0.171; 16.9) |
| 25W | 40 | 18.6 | 0.443 (0.191; 20.7) | 219 | 99.5 | 13.6 (0.231; 52.94) | | | ND |
| **LNG** | | | | | | | | | |
| D0 | 16 | 7.44 | 1.14 (0.339; 5.11) | 14 | 6.36 | 1.26 (0.432; 6.24) | 30 | 6.90 | 1.14 (0.339; 6.24) |
| 25W | 9 | 4.19 | 1.01 (0.368; 5.90) | 6 | 2.73 | 0.613 (0.486; 7.08) | 15 | 3.45 | 0.896 (0.368; 7.08) |
| **ETG** | | | | | | | | | |
| D0 | 1 | 0.465 | 1.37 (1.37; 1.37) | 4 | 1.82 | 0.667 (0.508; 1.84) | 5 | 1.15 | 0.712 (0.508; 1.84) |
| 25W | 2 | 0.930 | 0.673 (0.545; 0.0801) | 1 | 0.455 | 1.02 (1.02; 1.02) | 3 | 0.700 | 0.801 (0.545; 1.02) |
| **NES** | | | | | | | | | |
| D0 | 2 | 0.930 | 0.428 (0.229; 0.626) | 6 | 2.73 | 0.248 (0.140; 2.08) | 8 | 1.84 | 0.260 (0.140; 2.08) |
| 25W | 7 | 3.31 | 0.291 (0.148; 0.864) | 7 | 3.18 | 0.329 (0.143; 1.90) | 14 | 3.22 | 0.310 (0.143; 1.90) |
| **GES** | | | | | | | | | |
| D0 | 0 | 0.0 | 0.00 (0.00; 0.00) | 2 | 0.909 | 2.16 (1.77; 2.54) | 2 | 0.460 | 2.16 (1.77; 2.54) |
| 25W | 1 | 0.465 | 4.96 (4.96; 4.96) | 1 | 0.455 | 3.84 (3.84; 3.84) | 2 | 0.460 | 4.40 (3.84; 4.96) |

\* n-values given are the number of participants that had specified progestin at concentrations > LLOQ in either the DMPA-IM arm (n = 215 total), the NET-EN arm (n = 220 total) or from the whole cohort (n = 435 total)

# % refers to the percentage of participants that had specified progestin at concentrations > LLOQ in the respective arms or from the whole cohort. Median (Min; Max) refers to the median value with Min in brackets being the lowest concentration of specified progestin measured in the participants that is > LLOQ and Max in brackets being the highest concentration of specified progestin measured in the participants. ND–Values not determined at 25W for study progestin (MPA or NET in whole cohort).

individuals that had quantifiable levels (> LLOQ) of any non-study progestin at 25W, as well as all those that missed their study progestin injection at 24W (as indicated in Methods). A total of 119 women (54 in the DMPA-IM arm and 65 in the NET-EN arm) were excluded in the subgroup PP1 analysis. When comparing the demographic characteristics of the women between arms for the PP1 subgroup, no significant differences were detected (Table 1).

In the DMPA-IM arm, the median concentration of MPA at 25W was 6.83 nM (2.64 ng/mL) (IQR 5.09–8.80 nM), while in the NET-EN arm the median concentration of NET at 25W was 13.3 nM (3.97 ng/mL) (IQR 9.28–18.5 nM) in the PP1 analysis (Table 4), which is very similar to the values obtained for the whole cohort in the mITT analysis (Table 2). The mean concentrations of MPA and NET at 25W, as well as the highest individual MPA and NET concentrations, in the DMPA-IM and NET-EN arms, respectively (S7 Table), are also similar to those obtained at 25W in the mITT analysis (S4 Table).

Both arms exhibited a wide inter-individual range of concentrations of the assigned progestin at 25W (0.168–20.4 nM for MPA in the DMPA-IM arm and 2.42–52.9 nM for NET in the NET-EN arm) of the subgroup PP1 analysis (S7 Table). Levene's test for differences in variance, indicated that there was more inter-individual variation in the NET-EN arm compared to the DMPA-IM arm ($p < 0.001$) (data not shown).

**Table 4. Analysis of imputed concentrations of MPA and NET for subgroup of whole cohort (PP1).**

| | DMPA-IM | | NET-EN | | DMPA-IM vs NET-EN |
|---|---|---|---|---|---|
| | Median (IQR) | n | Median (IQR) | n | p-value* |
| **MPA (nM)** | | | | | |
| D0 | 0.251 (0.00; 1.35) | 161 | 0.0640 (0.00; 0.732) | 155 | **0.008** |
| 25W | 6.83 (5.09; 8.80) | 161 | 0.00 (0.00; 0.0640) | 155 | **<0.0001** |
| Change (25W - D0) | 5.83 (3.93; 8.18) | | -0.0640 (-0.719; 0.00) | | **<0.0001** |
| Change p-value* | **<0.0001** | | **<0.0001** | | |
| **NET (nM)** | | | | | |
| D0 | 0.00 (0.00; 0.0840) | 161 | 0.00 (0.00; 0.0840) | 155 | 0.929 |
| 25W | 0.00 (0.00; 0.00) | 161 | 13.3 (9.28; 18.5) | 155 | **<0.0001** |
| Change (25W - D0) | 0.00 (-0.0840; 0.00) | | 13.1 (8.91; 18.5) | | **<0.0001** |
| Change p-value* | **<0.0001** | | **<0.0001** | | |

*Unadjusted; Within group analysis—Wilcoxon matched-pairs signed rank test; Between groups (D0 vs D0 and 25W vs 25W) and Change (absolute 25W –D0)—Mann Whitney test.

## High baseline and 24W weight and BMI associate with serum MPA and NET concentrations

Most of the women were either overweight (DMPA-IM 23.5% and NET-EN 22.9%) or obese (DMPA-IM 47.5% and NET-EN 39.0%) at D0. A significant difference between the DMPA-IM and NET-EN arms was detected at D0 for weight (kg) and BMI in the mITT analysis (Table 1).

To investigate possible associations between progestin concentrations and baseline weight and BMI, we excluded all participants with non-study progestin concentrations greater than 1.5 nM at 25W or at D0, as well as all those that missed their study progestin injection at 24W (as indicated in Methods) (PP2 analysis). For this PP2 subgroup, we did not detect significant differences between the DMPA-IM and NET-EN arms at baseline for weight (p = 0.22) or BMI (p = 0.12). We detected significant negative correlations between the 25W MPA concentrations and D0 weight and BMI in the DMPA-IM arm (Fig 3 and Table 5). Similar significant negative correlations were obtained for NET at 25W and D0 weight and BMI in the NET-EN arm (Fig 3 and Table 5). Significant correlations between 25W MPA and NET concentrations with 24W weight and BMI was also detected (Table 5). Significant negative time-varying associations were detected between MPA and NET concentrations and D0 weight and D0 BMI (Change 25W - D0 MPA vs D0 weight/D0 BMI) in both the DMPA-IM and NET-EN arms, respectively, in this PP2 subgroup (S8 Table). When the women were grouped according to BMI, significant differences in median peak serum MPA and NET concentrations were detected between normal weight (MPA 7.32 and NET 16.0 nM) and obese (MPA 4.74 and NET 11.5 nM) women in the DMPA-IM and NET-EN arms, respectively (Fig 3 and Table 6). No significant differences in peak serum MPA or NET concentrations were detected when comparing normal to underweight or overweight women. Progestin comparison results were not affected by site (data not shown).

## Discussion

Ours is the first study to determine baseline and peak serum concentrations of MPA or NET from a relatively large cohort of women randomized to DMPA-IM (n = 215) or NET-EN (n = 220). This is the most robust data to date on these levels for each contraceptive as well as their comparative values in a head-to-head comparison. We report for the first time the

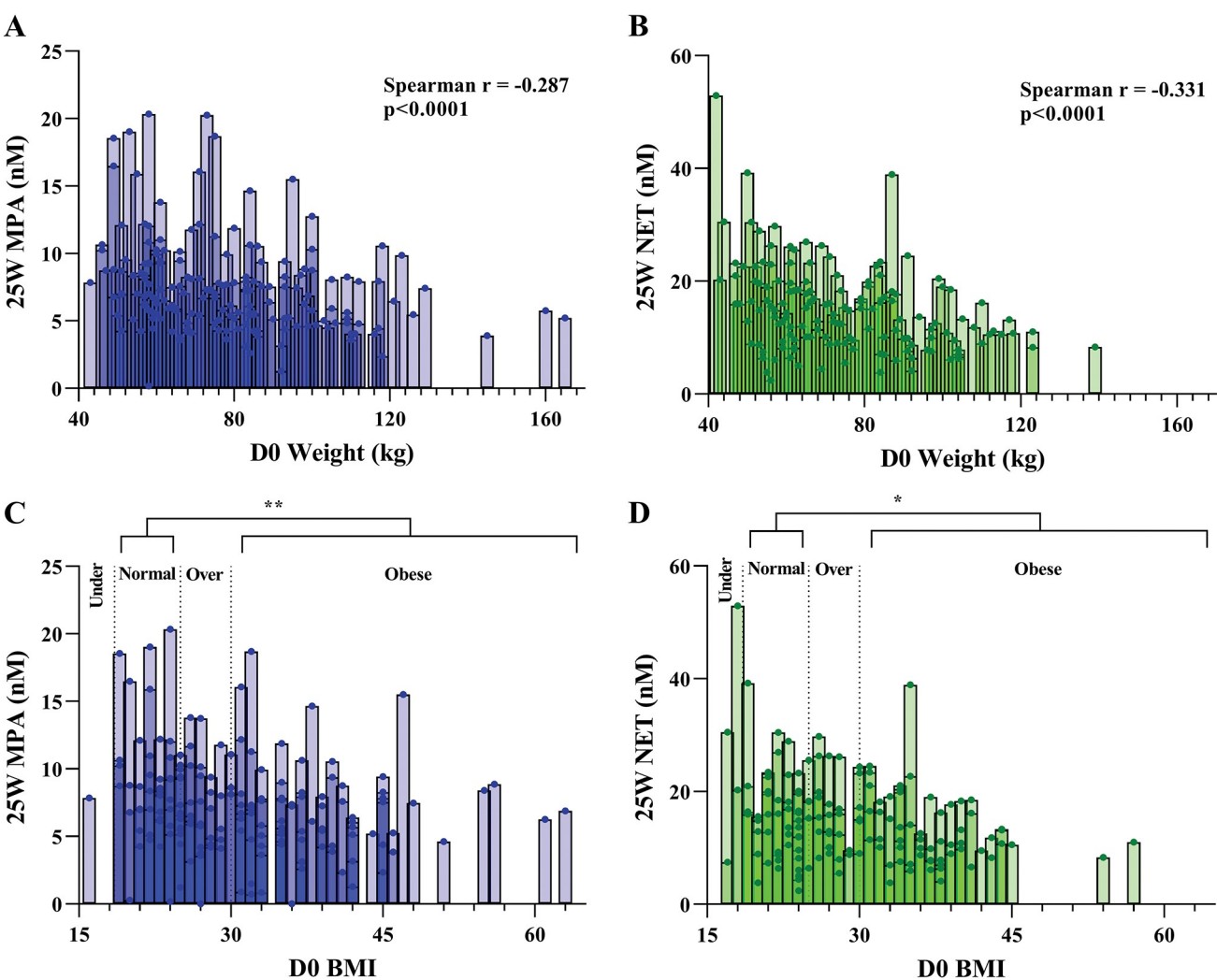

**Fig 3. Baseline (D0) weight and BMI influence peak serum (25W) MPA and NET concentrations after excluding non-study progestins at D0 and 25W in PP2 subgroup.** Graphs indicate D0 weight (kg) (A & B) or D0 BMI (C & D) plotted against 25W serum concentrations of MPA (nM) (A & C) or NET (nM) (B & D) in the DMPA-IM or NET-EN arm, respectively, for each individual. Non-parametric Spearman correlation analysis was performed (A–D). For (C & D) Kruskal-Wallis with Dunn's multiple comparison test was performed to compare 25W peak serum concentrations in normal weight women to those in other weight categories. Significant differences are indicated by asterisks where * and *** represent p<0.05 and p<0.001, respectively. Under—underweight; Over—overweight.

concentrations of study and non-study progestins measured for this relatively large cohort of women at baseline (D0) and at one week after a six-month injection (25W) to allow robust assessment of non-study progestins in a randomized trial at both sampling times, as well as subgroup analysis after exclusion of some participants with non-study or no study progestins at 25W. This comprehensive dataset allowed us to validate our prediction of peak study progestin serum levels occurring at one week after injection, as well as to detect significant associations within and between contraceptive arms, between progestin concentrations and BMI and weight for participants.

We report a median peak serum concentration of MPA for DMPA-IM users at 25W of 6.59 nM (2.55 ng/mL) (IQR 4.80–8.70 nM) for the whole cohort of 215 women (mITT group). A median peak serum MPA concentration of 6.83 nM (2.64 ng/mL) (IQR 5.09–8.80 nM) was obtained for the subgroup of 161 women (PP1 group) after excluding some participants with

**Table 5. Correlation analysis of 25W peak serum MPA and NET concentrations with weight (kg) and BMI in the DMPA-IM and NET-EN arms, respectively, after excluding non-study progestins at D0 and 25W (PP2).**

| | MPA 25W (nM)* | | NET 25W (nM)# | |
|---|---|---|---|---|
| | Spearman r | p-value | Spearman r | p-value |
| **Weight, kgs** | | | | |
| D0 | -0.287 | <**0.0001** | -0.331 | <**0.0001** |
| 24W | -0.286 | <**0.0001** | -0.326 | <**0.0001** |
| **BMI** | | | | |
| D0 | -0.238 | **0.0012** | -0.276 | **0.0007** |
| 24W | -0.234 | **0.0015** | -0.273 | **0.0007** |

*In the DMPA-IM arm only

#In the NET-EN arm only; Non-parametric Spearman correlation analysis.

any quantifiable non-study progestins at 25W. Many different studies have previously investigated peak serum concentrations for MPA in DMPA-IM users, some having measured at multiple time points, and using different methods including radioimmunoassay and LC-MS/MS (reviewed in [5–7]). Our data fall within the lower range of peak serum values (2.6 nM to 30 nM) reported for most of the published studies but are much lower than the range of 60 nM to 100 nM reported in some other studies (reviewed in [5–7]). We report a median peak serum concentration for NET at 25W of 13.6 nM (4.06 ng/mL) (IQR 9.01–19.0 nM) for the whole cohort of 220 women and 13.3 nM (3.97 ng/mL) (IQR 9.28–18.5 nM) for a subgroup of 155 women (PP1 analysis). Limited studies have investigated NET serum concentrations for NET-EN. Our NET peak serum results are similar to some previous reports [45–48], but about two- to five-fold lower than most other reports that sampled within 2 weeks post-injection [5]. Although we did not set out to perform a detailed pharmacokinetic analysis, we show that the trial yielded sufficient variation in sampling time after the 24W injection to indicate that the median time after injection to reach peak serum levels for both DMPA-IM and NET-EN injection is seven days (S1 Fig).

When comparing our results for both contraceptive arms, we report that the peak serum concentration for MPA of 6.83 nM is about 2-fold lower than the 13.35 nM value for NET, which may be partly due to the 1.26-fold lower dose in mmoles for DMPA-IM (150 mg, 0.388

**Table 6. Median and mean peak serum MPA and NET concentrations in different BMI categories after excluding non-study progestins at D0 and 25W (PP2).**

| | D0 BMI category | | | |
|---|---|---|---|---|
| | Under | Normal | Over | Obese |
| **MPA peak serum*** | | | | |
| n-value | 1 | 53 | 40 | 89 |
| Median (IQR) | 7.84 (7.84; 7.84) | 7.32 (5.78; 9.97) | 6.73 (5.58; 9.30) | 4.74 (6.02; 7.94) |
| Mean (Min; Max) | 7.84 (7.84; 7.84) | 8.38 (7.34; 9.42) | 7.68 (6.47; 8.89) | 6.69 (6.08; 7.31) |
| **NET peak serum**** | | | | |
| n-value | 4 | 52 | 28 | 64 |
| Median (IQR) | 24.4 (10.7; 47.3) | 16.0 (9.45; 20.1) | 13.0 (9.60; 19.0) | 11.5 (8.54; 16.9) |
| Mean (Min; Max) | 27.8 (-2.80; 58.4) | 15.7 (13.6; 17.8) | 15.2 (12.3; 18.1) | 13.1 (11.7; 14.6) |

*In DMPA-IM arm only

**In NET-EN arm only; under-underweight; over-overweight.

mmoles) than NET-EN (200 mg, 0.487 mmoles). Levene's test for differences in variance indicated that the inter-individual variability of MPA concentration (median 6.83 nM and range 0.168 nM to 20.4 nM) in the DMPA-IM arm is less than that for NET concentrations (median 13.3 nM and range 2.42 nM to 52.9 nM) in the NET-EN arm in our subgroup. Our findings are also unlikely to be confounded by differences in baseline characteristics between contraceptive arms as we do not detect any significant differences between demographic factors at baseline for the subgroups (PP1 or PP2 analysis), and only detect significant differences at baseline for two factors in the whole cohort (Table 1). Our finding that peak serum levels of MPA are lower than those for NET is consistent with previous reports for relative progestin levels at other time points after injection [18, 33, 40]. However, our finding of less inter-individual variability for peak serum MPA compared to NET was unexpected since our review of published data indicated a greater inter-individual variability for MPA than NET in DMPA-IM and NET-EN users, respectively [5]. Reasons for apparent discrepancies between our results and those reported by others for DMPA-IM and NET-EN may be due to differences in methods of detection, numbers of trial participants, sampling times, intrinsic participant demographic characteristics and confounders inherent in non-randomized trials.

It is important to consider the implications of our peak serum progestin concentration results regarding steroid receptor effects. Both MPA and NET concentrations are clearly sufficient to result in PR-mediated responses since both DMPA-IM and NET-EN use result in efficient contraceptive effects which are due to their progestogenic actions via the PR. What is more difficult to assess is whether peak serum MPA at around 6.59 nM will result in GR-mediated side-effects *in vivo*. The affinities of MPA and NET for the GR suggest that substantial GR saturation could occur at the peak serum concentrations of MPA but not NET in these contraceptive users. We have previously shown that NET, unlike MPA, has very little activity via the GR at concentrations as high as 100 nM [49–52], consistent with their affinities for the GR of 215 nM and 10.8 nM [53], respectively. Sensitivity to GR responses is highly dependent on GR levels [54] and we have also previously reported that the effects of MPA via the GR are ligand-, dose-, gene- and cell/tissue-specific *in vitro* and *ex vivo* [49–52]. Of particular relevance to side-effects mediated via the GR, we previously showed that unlike for NET, MPA increases HIV infection in PBMCs at 50 nM but not at 10 nM *ex vivo* [51] and increases HIV replication at 10 nM and 100 nM in endocervical explants, but only at 100 nM, and not at 10 nM, in ectocervical tissue explants [50], while others have detected significant effects of MPA on female genital tract primary cells *ex vivo* with 1 nM MPA [55]. Our data reveal lower peak serum levels for MPA and lower inter-individual variability for both MPA and NET, as well as the absence of outliers with MPA concentrations above 50 nM, as compared to several previous reports. Taken together and considering that *in vitro* and *ex vivo* results may not always mimic what happens *in vivo*, the median peak serum concentration of MPA reported in the current study may be below that required to result in substantial systemic GR-mediated effects for most, but not all genes and cells/tissues. However, it is possible that GR-mediated side-effects by MPA may occur in select tissues and cells, in particular in the female genital tract and where progestin concentrations and/or GR levels may differ from systemic levels.

MPA was the most commonly quantifiable non-study progestin at initiation in both arms (quantifiable in 54.3% of women) and at 25W in the NET-EN arm (quantifiable in 27.3% of women), followed by NET at D0 in both arms (28.5% of women) and at 25W in the DMPA-IM arm (18.6% of women), and then LNG for both arms at D0 (6.90% of women) and at 25W (3.40% of women) (Table 3). Whether the detection of these non-study progestins is due to slow metabolism and release resulting in hormone 'tails' or to active continued use after initiation is difficult to determine. Using predictions made by the available modelling data for DMPA-IM [35] and the limited information available for NET-EN [56], we estimate that

about 28.7% (MPA) and 21.8% (NET) of all the women in the cohort had non-study hormone 'tails' at initiation (S9 Table). We further estimate that 25.5% and 6.67% of all the women in the cohort misreported DMPA-IM and NET-EN usage, six and four months before initiation, respectively. In addition, we estimate that at 25W, 8.18% and 8.37% of the women in the NET-EN and DMPA-IM, respectively, had a hormone 'tail' due to at least one DMPA-IM or NET-EN injection six or four months before initiation, respectively. At 25W it is more difficult to discriminate between active use of non-study DMPA-IM or NET-EN after initiation and a hormone 'tail' due to misreporting prior use before initiation. Regarding LNG, since the women were not excluded if they were taking LNG-containing contraceptives before initiation, this could be the cause of quantifiable LNG at D0, as found in 30/435 (6.90%) women at D0 (in both arms). However, where levels of LNG at 25W were above 2 000 pg/mL (6.4 nM) [29], we estimate that only 1/435 (0.230%) of the women from both arms was engaged in ongoing use of LNG-containing contraceptives sometime during the study. ETG, NES and GES were detected in very few women at D0 or 25W, consistent with relatively infrequent use of these methods at the study sites.

Our results regarding the presence of non-study progestins are in accordance with a previous study which found that among 358 women reporting different forms of hormonal contraceptive use, including DMPA-IM and NET-EN, 28% of samples were not fully concordant with self-report, as were 14% of samples from 744 women reporting no hormonal contraceptive use [31]. Interestingly, these authors also found that MPA and LNG were the progestins most frequently detected in self-reported non-users [31]. Another study by Achilles et al. (Zim CHIC study) measured the presence of quantifiable non-study progestins LNG, ETG, NET and MPA, at baseline and at 30, 90 and 180 days after initiation in a non-randomized study with 5 arms, including a DMPA-IM and a NET-EN arm, performed in Zimbabwe [29]. Contrary to their results which reported only 4.5% of women having quantifiable MPA at baseline, we found 54.3% of women had quantifiable MPA at baseline. We also found many more women with quantifiable NET at baseline (28.5 vs < 0.5%) and much fewer women with quantifiable LNG at baseline (6.90 vs 23%). Our data are consistent with baseline information indicating contraceptive availability and/or preference at these sites in South Africa, where 74% versus 31% of women in the WHICH trial self-reported some previous use of DMPA-IM or NET-EN, prior to six or four months, respectively, before initiation (Table 1). We also found many more women (27.3%) with quantifiable non-study MPA at 25W (about 190 days) compared to the Achilles et al. study at 180 days (< 1%), and non-study NET at 25W (about 190 days) (18.6%) compared to the Achilles et al. study at 180 days (< 0.5%). Reasons for these differences between our results and those from the Zim CHIC study are likely due to regional preferences for contraception use, as well as that in the Zim CHIC study, women were excluded on the basis of self-reported use of any hormonal contraception < 30 days prior to initiation or DMPA-IM use < 10 months prior to initiation.

We show for the first time in a highly powered randomized study that the greater the BMI and weight at D0, the lower the progestin serum concentration at 25W, as well as the smaller the change from D0. Our data are consistent with findings from progestin-only LNG and ETG implant studies [13, 14]. It should be noted that most of the women in the WHICH trial were overweight or obese. This is consistent with data from South Africa showing that about 68% of women over the age of 15 are overweight or obese [2]. Our results suggest that at least part of the reason that our peak serum levels are lower than in some reports, for example in trials conducted among Thai women, is that they may be due to differences in BMI and weight of trial participants. The peak serum levels of MPA and NET in the DMPA-IM and NET-EN arms, respectively, for obese women were significantly lower (MPA about 1.54-fold, NET about 1.39-fold) than those of normal weight women. Our results suggest that side-effects dependent

on higher peak serum levels may be lower in overweight and obese women, and that modulation of contraceptive progestin dose could be considered depending on these factors.

Associations between contraceptive use and health outcomes or changes in biological markers from observational clinical trials are likely to be confounded by multiple factors, including differences at baseline in demographic factors, especially BMI and weight, numbers of participants misreporting contraceptive use, a lack of objective validation of progestin levels at initiation and during the study, variations in sampling time and the use of inter-study comparisons. Randomized trials comparing contraceptives likely avoid some but not all of these potentially confounding factors. To the best of our knowledge, all published health outcomes and biological marker data thus far from clinical trials on contraceptives have not taken into account the possible confounder of non-study progestins at initiation and during follow-up when drawing conclusions, with the notable exception of data from the non-randomized Zim CHIC study [29, 33, 57]. In the ECHO trial [35], where 73.7% (5769/7830) of the participants were recruited from South Africa, concentrations of MPA (but not for other study or non-study progestins) in blood were measured for a sample of 7% of participants at initiation, but not reportedly after initiation. The ECHO trial reported that 13% of women had more than 1.04 nM MPA in the blood at initiation, compared to our results of 31.3%. The reasons for this large difference at D0 between the ECHO and WHICH trials are difficult to understand but may be due to different contraception use at sites sampled at D0 in the ECHO trial.

## Conclusions

Peak serum levels of MPA and NET are lower and less variable than expected from the literature, suggesting that side-effect profiles for both may be lower than predicted based only on progestin concentrations, and less variable between women. Most participants in the WHICH trial were overweight or obese, which we show for the first time significantly and negatively associates with the peak serum progestin levels for both DMPA-IM and NET-EN users. Based on affinities for the GR, MPA peak serum concentrations may exert side-effects via the GR in select tissues and on select genes, unlike NET. Our results suggest that when comparing effects of DMPA-IM in clinical trials in sub-Saharan Africa with those of other contraceptives that are much less widely used, analysis of health outcomes and biological responses are more likely to be confounded by the presence of MPA in the other contraceptive arms than the reciprocal effect. In such trials, effects of DMPA-IM relative to baseline are also likely underestimated in the DMPA-IM arm at sites where DMPA-IM is widely used. Biological and health outcomes may also be confounded by non-study progestins other than MPA, such as NET and LNG. This is likely due to high levels of misreporting of contraceptive use before and during trials. Furthermore, our study highlights the potential problem of progestin 'tails' for MPA and NET and suggests that not using DMPA-IM or NET-EN less than six or four months, respectively, prior to initiation is insufficient time to control for this potential problem. In addition, reported side-effect and health outcome profiles are likely to differ between trials where median BMIs and weights of participants are significantly different.

## Supporting information

**S1 Fig. Peak serum MPA and NET concentrations are highest in participants seven days after the last DMPA-IM or NET-EN injection (mITT group).** Graphs indicate days since the last DMPA-IM (A) or NET-EN (B) injection at 24 weeks (24W) and when blood was collected for determination of serum MPA or NET concentrations plotted against the determined serum MPA (nM) (A) or NET (nM) (B) concentrations in the DMPA-IM or NET-EN arms, respectively, for each participant. Non-parametric Spearman correlation analysis was

performed and Spearman r- and p-values for the correlations are given.
(TIF)

**S1 Table. Nomenclature, retention times, quantifier and qualifier mass transitions, collision energies and cone voltages of target analytes and internal standards.**
(DOCX)

**S2 Table. Comprehensive method validation data\*.**
(DOCX)

**S3 Table. Limit of Detection (LOD), lower and upper limit of quantification (LLOQ and ULOQ, respectively)\*.**
(DOCX)

**S4 Table. Mean and range of imputed MPA and NET concentrations for whole cohort (mITT).**
(DOCX)

**S5 Table. Time (days) between final injection (24W) and sampling time.**
(DOCX)

**S6 Table. Non-study contraceptive use per study site for the whole cohort (mITT).**
(DOCX)

**S7 Table. Mean and range of imputed MPA and NET concentrations analyzed for subgroup of the whole cohort (PP1).**
(DOCX)

**S8 Table. Time varying associations between the change (25W-D0) in MPA or NET concentrations with change in baseline weight or BMI.**
(DOCX)

**S9 Table. Estimated percentage of women misreporting DMPA-IM or NET-EN use or possible 'tail' effects.**
(DOCX)

**S1 Protocol. WHICH clinical trial additional methods.**
(DOCX)

**S2 Protocol. UHPLC-MS/MS progestin quantification methods.**
(DOCX)

## Acknowledgments

We thank all the participants who participated in the WHICH clinical trial. We thank Karen van der Merwe for project administration and assistance in preparation and submission of the manuscript. We thank Marietjie Stander and Erick van Schalkwyk for assistance with the UHPLC-MS/MS assays.

## Author Contributions

**Conceptualization:** Chanel Avenant, Janet P. Hapgood.

**Data curation:** Chanel Avenant, Yusentha Balakrishna, Ishen Seocharan.

**Formal analysis:** Chanel Avenant, Yusentha Balakrishna, Pai-Lien Chen.

**Funding acquisition:** Mandisa Singata-Madliki, Janet P. Hapgood.

**Investigation:** Chanel Avenant, Alexis J. Bick, Sigcinile Dlamini, Johnson Mosoko Moliki.

**Methodology:** Chanel Avenant, Salndave B. Skosana, Karl-Heinz Storbeck, Janet P. Hapgood.

**Project administration:** Chanel Avenant, Janet P. Hapgood.

**Resources:** Mandisa Singata-Madliki, Jenni Smit, Mags Beksinska, Ivana Beesham, Joanne Batting.

**Supervision:** Chanel Avenant, Janet P. Hapgood.

**Visualization:** Chanel Avenant, Yusentha Balakrishna.

**Writing – original draft:** Chanel Avenant, Alexis J. Bick, Janet P. Hapgood.

**Writing – review & editing:** Chanel Avenant, Alexis J. Bick, Sigcinile Dlamini, Yusentha Balakrishna, Johnson Mosoko Moliki, Mandisa Singata-Madliki, G. Justus Hofmeyr, Jenni Smit, Mags Beksinska, Ivana Beesham, Ishen Seocharan, Joanne Batting, Pai-Lien Chen, Karl-Heinz Storbeck, Donita Africander, Janet P. Hapgood.

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
