## [Decision Letter · Decision Letter 0]

22 Aug 2023

PONE-D-23-20376Misreporting contraceptive use and the association of peak study progestin levels with weight and BMI among women randomized to the progestin-only injectable contraceptives DMPA-IM and NET-EN.PLOS ONE

Dear Dr. Hapgood,

Thank you for submitting your manuscript to PLOS ONE. After careful consideration, we feel that it has merit but does not fully meet PLOS ONE’s publication criteria as it currently stands. Therefore, we invite you to submit a revised version of the manuscript that addresses the points raised during the review process.

ACADEMIC EDITOR: Read through the manuscript to correct all the errors  

We look forward to receiving your revised manuscript.

Kind regards,

Andrew Max Abaasa, Ph.D.

Academic Editor

PLOS ONE

Additional Editor Comments:

Read through the manuscript sentence by sentence to correct a few errors

Reviewers' comments:

Reviewer's Responses to Questions

**Comments to the Author**

1. Is the manuscript technically sound, and do the data support the conclusions?

Reviewer #1: Yes

Reviewer #2: Yes

2. Has the statistical analysis been performed appropriately and rigorously? 

Reviewer #1: Yes

Reviewer #2: Yes

3. Have the authors made all data underlying the findings in their manuscript fully available?

Reviewer #1: Yes

Reviewer #2: Yes

4. Is the manuscript presented in an intelligible fashion and written in standard English?

Reviewer #1: Yes

Reviewer #2: Yes

5. Review Comments to the Author

Reviewer #1: Thank you for the write up on a very important topic of hormonal contraception and side effects. The comparison of peak levels of hormone versus BMI /weight changes plus the confounding of non study hormonal levels makes very interesting findings. Great write up!

Reviewer #2: This study evaluated baseline and peak (25wk) serum concentrations of MPA and NET-EN in a randomized trial of these two injectables, as well as baseline and 25wk concentrations of non-study contraceptive hormones. The authors found a large proportion of women that had detectable and/or high levels of off-study progestin levels even at 25 weeks which indicates either a long tail from pre-study use or misreporting of continued contraceptive use. The authors also verified that 7 days post injection was the appropriate time point to measure peak serum concentrations which is important information for the design of future studies. A negative correlation between weight/BMI and peak MPA and NET-EN concentrations was also observed in a subset of women which may explain some inter-study variability of results. I have a couple of comments.

In the methods, for the PP1 and PP2 cohorts, please list the number of women excluded for missed 24w injection and for presence of non-study progestins in each group. It appears that a third of the NET EN group were excluded for both the PP1 and PP2 analyses so it would be important to understand if it is due to non-compliance with the injection schedule or presence of non-study progestins. However, only a quarter of the women in the MPA group were excluded for PP1 and 16% for PP2 so it appears that these women were more compliant with the 24w injection.

Table 6. The footnote symbols are not represented within the tables.

6. PLOS authors have the option to publish the peer review history of their article (what does this mean?). If published, this will include your full peer review and any attached files.

Reviewer #1: No

Reviewer #2: No

---

## [Decision Letter · Decision Letter 1]

4 Dec 2023

Misreporting contraceptive use and the association of peak study progestin levels with weight and BMI among women randomized to the progestin-only injectable contraceptives DMPA-IM and NET-EN.

PONE-D-23-20376R1

Dear Dr.Janet P Hapgood

We’re pleased to inform you that your manuscript has been judged scientifically suitable for publication and will be formally accepted for publication once it meets all outstanding technical requirements.

Kind regards,

Fabio Vasconcellos Comim, MD,PhD

Academic Editor

PLOS ONE

**Comments to the Author**

Reviewer #2: All comments have been addressed

2. Is the manuscript technically sound, and do the data support the conclusions?

Reviewer #2: Yes

3. Has the statistical analysis been performed appropriately and rigorously? 

Reviewer #2: Yes

4. Have the authors made all data underlying the findings in their manuscript fully available?

Reviewer #2: Yes

5. Is the manuscript presented in an intelligible fashion and written in standard English?

Reviewer #2: Yes

6. Review Comments to the Author

Reviewer #2: (No Response)

7. PLOS authors have the option to publish the peer review history of their article (what does this mean?). If published, this will include your full peer review and any attached files.

Reviewer #2: No

---

## [Editor Report · Acceptance letter]

14 Dec 2023

PONE-D-23-20376R1 

PLOS ONE

Dear Dr. Hapgood, 

I'm pleased to inform you that your manuscript has been deemed suitable for publication in PLOS ONE. Congratulations! Your manuscript is now being handed over to our production team.

Kind regards, 

on behalf of

Prof Fabio Vasconcellos Comim 

Academic Editor

PLOS ONE